# Independent Correlates of Glycemic Control among Adults with Diabetes in South Africa

**DOI:** 10.3390/ijerph21040486

**Published:** 2024-04-16

**Authors:** Abdulaziz Hamid, Aprill Z. Dawson, Yilin Xu, Leonard E. Egede

**Affiliations:** 1Department of Medicine, Medical School, Medical College of Wisconsin, Milwaukee, WI 53226, USA; 2Department of Medicine, Division of General Internal Medicine, Medical College of Wisconsin, 8701 Watertown Plank Road, Milwaukee, WI 53226, USA; adawson@mcw.edu; 3Center for Advancing Population Science, Medical College of Wisconsin, Milwaukee, WI 53226, USA; yilxu@mcw.edu

**Keywords:** diabetes, glycemic control, South Africa, sub-Saharan Africa, hemoglobin A1c

## Abstract

Background: Globally, the prevalence of diabetes is increasing, especially in low- and middle-income countries (LMICs), including those in the sub-Saharan African region. However, the independent socioeconomic correlates of glycemic control as measured by hemoglobin A1C have yet to be identified. Therefore, the aim of this analysis was to understand the independent correlates of glycemic control in South Africa. Methods: Data from the 2016 South Africa Demographic and Health Survey on adults with diabetes were used for this analysis. The dependent variable, glycemic control, was defined using hemoglobin A1c (HbA1c). Independent variables included: age, gender, ethnicity, marital status, region, urban/rural residence, ability to read, education, insurance, wealth, occupation, and employment in the last year. Analysis of variance was used to test for differences in mean HbA1c for each category of all independent variables, and a fully adjusted linear regression model was used to identify independent correlates of glycemic control (HbA1c). Results: Among the 772 people included in this analysis, there were significant differences in mean HbA1c by age (*p* < 0.001), ethnicity (*p* < 0.001), place of residence (*p* = 0.024), wealth index (*p* = 0.001), and employment in the last year (*p* = 0.008). Independent correlates of HbA1c included age, ethnicity, and wealth index. Conclusions: This study used data from a large diverse population with a high prevalence of diabetes in sub-Saharan Africa and provides new evidence on the correlates of glycemic control and potential targets for interventions designed to lower HbA1c and improve diabetes-related health outcomes of adults in South Africa.

## 1. Introduction

Globally, the prevalence of diabetes is increasing, especially in low- and middle-income countries (LMICs), including those in the sub-Saharan African region [1]. In sub-Saharan Africa, the prevalence of diabetes increased from 3.8% in 1980 to 8.7% in 2014 [2]. The cost of diabetes in sub-Saharan Africa was approximately 19.45 billion USD in 2015, accounting for 1.2% of total GDP, and is expected to increase to 59 billion USD by 2030 [3,4]. South Africa, a sub-Saharan African country, has the highest prevalence of diabetes in Africa, with rates increasing from 7% in 2010 to almost 13% in 2019 [1]. The estimated cost of diabetes in this country was over 198 million USD in 2018 [3]. In addition to being costly, individuals with diabetes are more likely to suffer from complications including blindness, heart disease, stroke, chronic kidney disease, and lower-extremity amputation [5]. However, among individuals with diabetes, those with good glycemic control have a lower risk of having diabetes-related complications compared to those with poor glycemic control [6].

Glycemic control, one of the key measures of diabetes management, is defined as a hemoglobin A1c (HbA1c) level of less than 7% [7]. Good glycemic control has been shown to be associated with a 13% reduction in eye complications, such as retinopathy, and a 20% reduction in kidney complications, including end-stage kidney disease and nephropathy [8]. Alternatively, poor glycemic control is linked to higher overall health care costs and poor physical health outcomes, such as retinopathy, heart disease, and chronic renal disease [1,3]. The financial burden of diabetes and complications can be mitigated through preventative measures and good glycemic control [9], which may be achieved by addressing social determinants of health [10,11].

South Africa is a large, diverse country of over 58 million people and multiple ethnic groups [12]. Of the over 58 million people in South Africa, about 81% are Black African, 8% Colored, 7% White, and 3% Indian/Asian [12]. Unfortunately, South Africa is plagued by the history of apartheid that has led to socioeconomic inequities, the perpetuation of poverty and inequitable access to quality health care, which have led to disparities in outcomes such as glycemic control [12,13,14]. Given the aging of the global population that is contributing to the increased prevalence of diabetes in this region, it is important to understand the specific social determinants and risk factors driving glycemic control in this environment. It has been well established that social determinants of health and social risk factors are key drivers of glycemic control [11,15,16], with social risks such as unemployment, lower income, lack of social support, lack of formal education, and food insecurity having been found to be associated with poor glycemic control in sub-Saharan African countries [17,18,19,20,21].

While there is literature on the drivers of glycemic control in a few sub-Saharan African countries, there is not a wide body of knowledge on the drivers of glycemic control in LMICs or in large diverse countries such as South Africa, where the highest prevalence of diabetes in sub-Saharan Africa is found [22]. As a result, it is of utmost importance to answer the question of what the socioeconomic correlates associated with glycemic control in South Africa, an LMIC in sub-Saharan Africa, are. By answering this important research question, targeted interventions can be developed to improve diabetes risk factor control and overall outcomes in this understudied area. Therefore, the aim of this analysis was to identify the correlates of glycemic control among adults with diabetes in South Africa using the (not widely available) biological measure of hemoglobin A1c.

## 2. Methods

### 2.1. Data Source and Study Population

This study is a secondary data analysis of publicly available international data that were collected by the Demographic and Health Survey (DHS) program through five surveys conducted in South Africa between June 2016 and November 2016, the most recent year of available data for South Africa. The DHS program collects data on population health from over 100 countries around the world. The primary goal of the South Africa Demographic and Health Survey 2016 (SADHS 2016) was to present current estimates of fundamental demographic and health indicators. The Statistics South Africa Master Sample Frame that was created by using enumeration areas from the 2011 census was used for the SADHS [23]. South Africa is divided into nine provinces, and the SADHS was designed to provide estimates for the entire country of South Africa. As a result, primary sampling units were allocated to ensure survey precision across the nine provinces (Western Cape, Eastern Cape, Northern Cape, Free State, Kwazulu-Natal, North West, Gauteng, Mpumalanga, and Limpopo) (Figure 1), and each province was stratified into urban, farm, and traditional areas, resulting in 26 sampling strata [23]. The SADHS used a stratified two-stage sample design, with stage 1 using a probability that was proportional to the size of primary sampling units and stage 2 of systematic sampling of dwelling units [23].

The number of dwelling units identified from the 2011 census was used as the primary sampling unit measure of size, with 750 primary sampling units selected from 26 sampling strata, which resulted in 468 primary sampling units in urban areas, 224 primary sampling units in traditional areas, and 58 primary sampling units in farm areas [23]. Lists of dwelling units served as the sampling frame, and a fixed number of 20 dwelling units per cluster were selected with systematic selection from the list [23]. Households within selected dwelling units were eligible for interviews and study participation. The SADHS is representative of the national population aged 15–49, and is representative at the national and provincial levels of both urban and nonurban areas [23]. Provinces containing small populations were oversampled. For example, the number of women interviewed ranged from 656 in Western Cape to 1360 in Kwazulu-Natal [23], and the total national sample size of women was 8514 before applying our specific study inclusion criteria. Similar methods were also used to sample men [23].

Training strategies used by the SADHS team included classroom training and field practice activities [23]. About 300 field workers (interviewers, supervisors, logistics officers, and nurses) were recruited and trained. The training course included instructions on interviewing techniques and administration of paper and electronic questionnaires, field procedures, questionnaire content review, and mock interviews [23]. More detailed information on the training procedures can be found in the SADHS Survey Methods manual [23].

The questionnaires used for the SADHS were based on the Demographic and Health Survey Program’s standard DHS questionnaires and were adapted to reflect the population and health issues that were relevant to South Africa. The DHS primarily collected nationally representative household and individual data through several questionnaires on population demographics, health, nutrition and wealth information. The household questionnaire contains information such as age, gender, education, type of place of residence, materials needed for security, and birth registration for both usual members of the household and visitors. The individual women’s and men’s questionnaires included topics such as background characteristics, reproductive behavior and intentions, employment, and gender roles. This study only includes adult women and men with diabetes in South Africa, and reports on the findings of an analysis that was conducted using previously collected SADHS data from 2016. The SADHS survey protocol was reviewed and approved by the South African Medical Research Council and Inner City Fund Institutional Review Board.

#### Diabetes Cohort

The information regarding whether a participant had been diagnosed with diabetes by a doctor or nurse was extracted from the Adult Health women’s and men’s questionnaire data files. Individuals who answered “yes” to the question of whether they had diabetes or had an HbA1c lab measurement greater than or equal to 6.5% were included in the analytic cohort. Out of the 15,292 households selected for interview, 11,083 were successfully interviewed. Among those households, there were 818 adults with diabetes, and out of the 818 adults with diabetes, 772 had hemoglobin A1c data avialable and were included in the analysis.

### 2.2. Dependent Variables and Correlates

#### 2.2.1. Dependent Variable—Glycemic Control 

Blood samples for SADHS study participants were collected by a nurse using a fingerstick and placing the blood on a filter paper card. The blood samples were dried overnight and transported to the Global Clinical and Viral Laboratory (GCVL) for further analysis. Hemoglobin A1c (HbA1c) was used to measure glycemic control for this study, and it was treated in the analysis as a continuous measure.

#### 2.2.2. Correlates—Socioeconomic Factors

Correlates included: age (15–21, 22–28, 29–38, or 39–59 years old), gender (female or male), ethnicity (Black/African, White, Colored, or Indian/Asian), marital status defined as married or living together (yes or no), region (Western Cape, Eastern Cape, Northern Cape, Free State, Kwazulu-Natal, North West, Gauteng, Mpumalanga, or Limpopo), place of residence (urban or rural), able to fully read (no or yes), educational level (none, primary school, secondary school, or higher education level), insurance (no or yes), wealth index (poorest, poorer, middle, richer, or richest), occupation group (not working, agricultural, clerical/sales, manual/domestic, or professional/technology/manger/service), and employment during the last 12 months (no or yes).

### 2.3. Statistical Analysis

Descriptive statistics were used to report frequencies and percentages, as well as means and standard deviations for all variables. Analysis of variance (ANOVA) was used to examine the difference in mean HbA1c for each category of all correlates. A fully adjusted linear regression model that included glycemic control as measured by continuous HbA1c as the outcome and the following independent variables was used to identify the independent correlates of glycemic control: age, gender, ethnicity, marital status, region, place of residence, able to fully read, educational level, insurance, wealth index, occupation group, and employment during the last 12 months. As the primary goal of the paper was to understand the independent socioeconomic correlates of glycemic control, variables were selected based primarily on the predisposing and enabling constructs of the Andersen model of health services utilization and outcomes research [24]. All analyses were performed using Stata v17.0, and statistical significance was defined as a *p*-value < 0.05.

## 3. Results

There were 772 adults in the SADHS living with diabetes included in this analysis. Almost half were aged 39–59 years (47.8%), and the majority were female (57.5%), Black/African (90.2%), not married (56.9%), did not have insurance (86.4%), were able to read (82.0%), and did not work in the last 12 months (54.9%). The mean HbA1c for the sample was 7.17 ± 1.89, and 77.3% had good glycemic control. Additional demographic and socioeconomic sample characteristics can be found in Table 1. Table 2 shows the mean HbA1c by demographic and socioeconomic characteristics. There were statistically significant differences in mean HbA1c by age (*p* < 0.001), ethnicity (*p* < 0.001), place of residence (*p* = 0.024), wealth index (*p* = 0.001), and whether they had worked in the last 12 months (*p* = 0.008). Table 3 shows results from the fully adjusted linear model and independent correlates of glycemic control measured by continuous HbA1c. These correlates included age, ethnicity, and wealth index. Adults aged 39–59 had significantly higher mean HbA1c (B: 0.67; 95% CI: 0.22, 1.12) compared to adults aged 15–21. Black/African (B: −3.24; 95% CI: −4.46, −2.03), White (B: −3.95; 95% CI: −5.42, −2.47), and Colored adults (B: −3.09; 95% CI: −4.43, −1.74) had significantly lower mean HbA1c compared to Indian/Asian adults. Adults in the middle wealth (B: 0.52; 95% CI: 0.11, 0.94) and richest (B: 0.66; 95% CI: 0.06, 1.26) wealth categories had significantly higher mean HbA1c compared to adults in the poorest wealth index category.

## 4. Discussion

This study adds a significant contribution to the literature on understanding correlates of glycemic control in LMICs, specifically the country of South Africa, in sub-Saharan Africa. While data on glycemic control as defined using HbA1c are limited in LMICs and sub-Saharan African countries, the DHS dataset for South Africa provides a unique opportunity to identify correlates of glycemic control measured by HbA1c among adults with diabetes. Results from this analysis showed almost one quarter of adults with diabetes in South Africa have poor glycemic control defined by a HbA1c level of greater than or equal to 7%. Findings also showed there were significant differences in mean HbA1c by age, ethnicity, place of residence, wealth index, and employment in the last year. In the fully adjusted linear regression model, independent correlates of glycemic control were identified as age, ethnicity and wealth index. Specifically, older adults, Indians/Asians, and those in the middle and richest wealth groups had significantly higher mean HbA1c compared to younger adults, all other ethnicities, and the poorest, respectively.

The country of South Africa has a large diverse population with a high prevalence (13%) of diabetes. This analysis showed Whites with diabetes had the lowest mean HbA1c (6.7%), followed by Black Africans (7.1%), Colored adults (7.8%), and Indian/Asian adults, with a striking mean HbA1c of 10.6%. In addition, all ethnic groups had significantly lower mean HbA1c compared to Indians/Asians by at least 3%. Tailored interventions that account for cultural differences and address the specific needs of the Indian/Asian, Colored, and Black African ethnic groups in South Africa need to be developed to improve glycemic control. Immigrants in South Africa face challenges in accessing health care services due to lack of documentation, limited resources, xenophobia, language and cultural differences, and a lack of clear understanding of the health care system [25]. In addition, Black Africans and Colored individuals are significantly more likely to obtain health care from the public health system, whereas Whites are more likely to obtain care from the private health system. In this two-part system, the free public system services about 84% of the population and the private sector services about 16% of the population [26]. However, about 73% of White individuals have access to private care compared to 52% of Indian/Asian individuals, 17% of Colored individuals, and only 10% of Black African individuals [26]. The public system, though commonly utilized by Black African, Colored, and Indian/Asian individuals, is overburdened, under-resourced, and plagued by long waiting times, adverse events, and other systemwide factors negatively impacting health outcomes [27,28]. Interventions designed to increase access to equitable health care and to shore up the public health system are needed to improve outcomes of adults with diabetes in South Africa.

While many studies commonly use income as a measure of economic status, we were able to use a more robust measure of wealth, which more accurately describes one’s financial status and access to assets in the context of LMICs and sub-Saharan Africa, facilitating cross-national comparisons. Consistent with the literature on LMICs, our findings showed individuals with greater wealth (middle wealth and richest) had higher mean HbA1c compared to the poorest individuals with diabetes in this analysis. Individuals with greater wealth in LMICs and specifically countries in sub-Saharan Africa have been found to have poorer health outcomes, including higher risk of hypertension and diabetes [29]. Interventions designed to provide continued diabetes self-management education and skills training for individuals with greater wealth are needed to improve glycemic control amongst South African Adults with diabetes.

This study provides new evidence on the correlates of glycemic control for South Africa, an LMIC in sub-Saharan Africa, using the biological measure of HbA1c. Given diabetes is under-recognized and under-diagnosed in a region of the world where exponential increases in the older adult population are expected to occur in the next two to three decades, and with this increase in older adult population an increase in noncommunicable diseases such as diabetes, it is imperative that these study findings that identify socioeconomic factors, which can serve as potential targets for interventions to improve glycemic control, be disseminated within international research, clinical, and political communities. Findings such as those identified in this study can be used to increase the development and testing of efficacious and effective interventions that will improve the health of individuals with diabetes. The findings of this study provide important clinical, research, and policy implications, given this new information on the independent correlates of glycemic control among adults with diabetes in South Africa. Conducting this study with data from South Africa, a country with a large diverse population and high prevalence of diabetes, provided a unique opportunity to understand the independent correlates of diabetes within an ethnically diverse low- and middle-income country (LMIC). In addition, study findings highlighted the ethnic disparities in glycemic control and illustrated the grave need for interventions for Black, Colored, and Indian/Asian adults with diabetes. Clinical implications of these study findings suggest health care providers of adults with diabetes in South Africa should focus on delivering additional diabetes education and continuous skill training for older adults, those in the middle and richest wealth group, and for Black Africans, Colored adults, and Indian/Asian adults. Findings from the study provide information on factors that are associated with increasing hemoglobin A1c, once again affording an opportunity to intervene earlier and before glycemic control reaches poor levels and increases one’s risk of diabetes-related complications in under-resourced areas that are not equipped to manage complications and complex cases. Therefore, the research implications highlight the importance of developing and testing novel interventions that target the cultural and lifestyle needs of adults in these groups. Finally, political implications include the need for increased access to equitable health care systems and dismantling of systemic factors perpetuating the negative impact of apartheid on socioeconomic status and health.

This study is strengthened by: (1) the use of data from a large, diverse, national sample of South African adults with diabetes, (2) the use of HbA1c as a measure of glycemic control, and (3) the inclusion of multiple socioeconomic factors to identify independent correlates of glycemic control. However, there are four noteworthy limitations. First, the study is cross-sectional, and thus the observed associations between demographic and socioeconomic factors and glycemic control do not imply causality. Second, diabetes status was based on self-report or HbA1c > 6.5%. As a result, individuals with both diagnosed and undiagnosed diabetes were included in the analysis. Third, only data for adults in South Africa were used for the analysis. As such, the findings may not be generalizable to other LMICs. Fourth, the analysis relied on self-reported measures of education and wealth, and while these variables may be subject to recall bias, self-reported measures of education and wealth are commonly used and widely accepted in health services research.

## 5. Conclusions

In this study of South African adults with diabetes, independent correlates of glycemic control included age, ethnicity, and wealth index. Specifically, older adults, Indians/Asians, and those in the middle and richest wealth groups had significantly higher mean HbA1c compared to younger adults, all other ethnicities, and the poorest, respectively. This study used data from a large diverse population with a high prevalence of diabetes in sub-Saharan Africa and provides new evidence on the correlates of glycemic control and potential targets for interventions designed to lower HbA1c and improve diabetes-related health outcomes of adults in South Africa.

## Figures and Tables

**Figure 1 ijerph-21-00486-f001:**
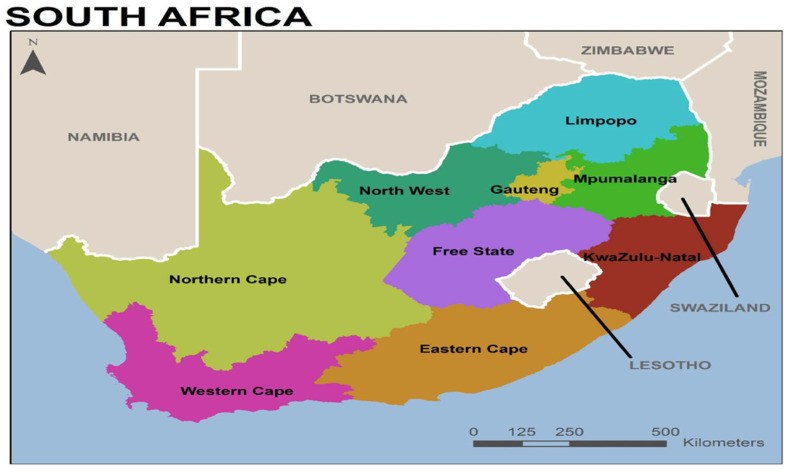
Map of South Africa and provinces from: National Department of Health (NDoH), 2019 [23].

**Table 1 ijerph-21-00486-t001:** Demographic and socioeconomic correlates of glycemic control in adults with diabetes in South Africa.

Variable	Frequency (%) Mean ± Standard Deviation n = 772
Age Group	
15–21 years	118 (15.3%)
22–28 years	95 (12.3%)
29–38 years	190 (24.6%)
39–59 years	369 (47.8%)
Gender	
Female	444 (57.5%)
Male	328 (42.5%)
Ethnicity	
Black/African	696 (90.2%)
White	17 (2.2%)
Colored	49 (6.4%)
Indian/Asian	10 (1.3%)
Educational Level	
None	35 (4.5%)
Primary	127 (16.5%)
Secondary	541 (70.1%)
Higher	69 (8.9%)
Married/Living Together	
No	439 (56.9%)
Yes	333 (43.1%)
Region	
Western Cape	37 (4.8%)
Eastern Cape	132 (17.1%)
Northern Cape	39 (5.1%)
Free State	102 (13.2%)
Kwazulu-Natal	122 (15.8%)
North West	67 (8.7%)
Gauteng	61 (7.9%)
Mpumalanga	102 (13.2%)
Limpopo	110 (14.3%)
Place of Residence	
Urban	366 (47.4%)
Rural	406 (52.6%)
Insurance Coverage	
No	667 (86.4%)
Yes	105 (13.6%)
Able to Fully Read	
No	139 (18.0%)
Yes	633 (82.0%)
Wealth Index	
Poorest	179 (23.2%)
Poorer	165 (21.4%)
Middle	190 (24.6%)
Richer	141 (18.3%)
Richest	97 (12.6%)
Occupation Group	
Not working	472 (61.1%)
Agriculture	18 (2.3%)
Clerical/Sales	34 (4.4%)
Manual/Domestic	143 (18.5%)
Professional/Technician/Manager/Service	105 (13.6%)
Worked Last 12 Months	
No	424 (54.9%)
Yes	348 (45.1%)
Poor Glycemic Control (HbA1c ≥ 7%)	
A1c < 7	597 (77.3%)
A1c ≥ 7	175 (22.7%)
Mean HbA1c	7.17 ± 1.89

**Table 2 ijerph-21-00486-t002:** Mean hemoglobin A1c by demographic characteristics.

Variable	Mean ± Standard Deviation	*p*-Value
Age Group		**<0.001**
15–21 years	**6.62 ± 0.52**	
22–28 years	**6.86 ± 1.45**	
29–38 years	**7.01 ± 1.49**	
39–59 years	**7.51 ± 2.34**	
Gender		0.617
Female	7.14 ± 1.81	
Male	7.21 ± 2.00	
Ethnicity		**<0.001**
Black/African	**7.09 ± 1.71**	
White	**6.65 ± 0.96**	
Colored	**7.75 ± 2.72**	
Indian/Asian	**10.63 ± 4.99**	
Educational Level		0.352
None	6.85 ± 0.63	
Primary	7.33 ± 2.10	
Secondary	7.13 ± 1.81	
Higher	7.40 ± 2.47	
Married/Living Together		0.075
No	7.07 ± 1.79	
Yes	7.31 ± 2.02	
Region		0.102
Western Cape	7.96 ± 2.99	
Eastern Cape	7.10 ± 1.63	
Northern Cape	7.66 ± 2.51	
Free State	7.16 ± 2.00	
Kwazulu-Natal	7.05 ± 1.56	
North West	7.06 ± 1.68	
Gauteng	7.50 ± 2.25	
Mpumalanga	6.98 ± 1.44	
Limpopo	7.04 ± 1.93	
Place of Residence		**0.024**
Urban	**7.33 ± 2.19**	
Rural	**7.03 ± 1.57**	
Insurance Coverage		0.417
No	7.15 ± 1.86	
Yes	7.31 ± 2.11	
Able to Fully Read		0.767
No	7.13 ± 1.75	
Yes	7.18 ± 1.92	
Wealth Index		**0.001**
Poorest	**6.78 ± 0.86**	
Poorer	**7.06 ± 1.88**	
Middle	**7.34 ± 2.19**	
Richer	**7.19 ± 1.89**	
Richest	**7.72 ± 2.47**	
Occupation Group		0.077
Not working	7.05 ± 1.71	
Agriculture	7.72 ± 4.23	
Clerical/Sales	7.42 ± 1.95	
Manual/Domestic	7.16 ± 1.72	
Professional/Technician/Manager/Service	7.56 ± 2.20	
Worked Last 12 Months		**0.008**
No	**7.01 ± 1.69**	
Yes	**7.37 ± 2.10**	

Bold = statistically significant.

**Table 3 ijerph-21-00486-t003:** Independent correlates of glycemic control.

Variable	Regression Coefficient (95% Confidence Interval)
Age Group	
15–21 years	Reference
22–28 years	0.11 (−0.40, 0.63)
29–38 years	0.24 (−0.24, 0.72)
39–59 years	**0.67 (0.22, 1.12) ****
Gender	
Female	Reference
Male	0.04 (−0.24, 0.32)
Ethnicity	
Indian/Asian	Reference
Black/African	**−3.24 (−4.46, −2.03) *****
White	**−3.95 (−5.42, −2.47) *****
Colored	**−3.09 (−4.43, −1.74) *****
Educational Level	
None	Reference
Primary	0.56 (−0.16, 1.27)
Secondary	0.33 (−0.40, 1.05)
Higher	0.30 (−0.59, 1.19)
Married/Living Together	
No	Reference
Yes	−0.05 (−0.36, 0.25)
Region	
Western Cape	Reference
Eastern Cape	−0.40 (−1.17, 0.37)
Northern Cape	−0.11 (−0.96, 0.73)
Free State	−0.55 (−1.34, 0.23)
Kwazulu-Natal	−0.67 (−1.47, 0.12)
North West	−0.65 (−1.49, 0.19)
Gauteng	−0.26 (−1.10, 0.57)
Mpumalanga	−0.57 (−1.37, 0.23)
Limpopo	−0.49 (−1.30, 0.31)
Place of Residence	
Urban	Reference
Rural	0.08 (−0.27, 0.42)
Insurance Coverage	
No	Reference
Yes	−0.31 (−0.76, 0.13)
Able to Fully Read	
No	Reference
Yes	0.09 (−0.33, 0.52)
Wealth Index	
Poorest	Reference
Poorer	0.25 (−0.15, 0.66)
Middle	**0.52 (0.11, 0.94) ***
Richer	0.35 (−0.13, 0.83)
Richest	**0.66 (0.06, 1.26) ***
Occupation Group	
Not working	Reference
Agriculture	0.34 (−0.67, 1.36)
Clerical/Sales	−0.14 (−0.97, 0.70)
Manual/Domestic	−0.40 (−1.01, 0.21)
Professional/Technician/Manager/Service	−0.13 (−0.78, 0.52)
Worked Last 12 Months	
No	Reference
Yes	0.43 (−0.15, 1.01)

Bold = statistically significant value; * *p*-value < 0.05; ** *p*-value < 0.01; *** *p*-value < 0.001.

## Data Availability

Restrictions apply to the availability of these data. Data were obtained from The Demographic and Health Surveys Program and are https://dhsprogram.com/data/available-datasets.cfm with the permission of The Demographic and Health Surveys Program.

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
