# Peer review of "Independent Correlates of Glycemic Control among Adults with Diabetes in South Africa"

_ijerph, 2024, doi:10.3390/ijerph21040486_

Round 1
Reviewer 1 Report
Comments and Suggestions for Authors
Type of manuscript: Article
Title: Independent Correlates of Glycemic Control Among Adults with Diabetes in South Africa
Manuscript ID: ijerph-2905738
Journal: IJERPH (ISSN 1660-4601)
Review Report: Thank you very much for providing me the opportunity to review this manuscript. While reading the manuscript, I could make the following comment from my side:
Summary: The escalating prevalence of diabetes, particularly in low and middle-income countries like those in Sub-Saharan Africa, underscores the urgent need for research on glycemic control. This analysis aimed to pinpoint the factors influencing glycemic control in South Africa. Utilizing data from the 2016 South Africa Demographic and Health Survey, variables such as age, ethnicity, and wealth index were examined alongside HbA1c levels. Significant disparities in HbA1c were observed across various demographics. This study, encompassing a sizable and diverse population, offers vital insights into enhancing interventions aimed at ameliorating diabetes-related health outcomes in South Africa.
Here are some general suggestions to the authors for conducting such study in future and the authors tried to follow a few of them conducting this research:
Sampling Method and Participant Recruitment: Determine how you will select participants, whether it's through random sampling, stratified sampling, or convenience sampling. Develop a plan to recruit participants from various regions in South Africa who have been diagnosed with diabetes. For sample size Calculation, determine the required sample size based on the expected effect size, power, and significance level.
Data Collection: Design a standardized questionnaire or interview protocol to collect relevant data from participants. This could include demographic information, medical history, lifestyle factors, and details about diabetes management. For Data Management, establish a system for data entry, storage, and management to ensure data integrity and security. Implement measures to ensure the quality and reliability of data collection, such as training interviewers and conducting pilot studies.
Dependent Variable: Glycemic control level (e.g., HbA1c levels or fasting blood glucose)
Independent Variables: Demographic factors (age, gender, ethnicity, socioeconomic status), Clinical factors (duration of diabetes, comorbidities, medication regimen), Lifestyle factors (diet, physical activity, smoking, alcohol consumption), Healthcare access and utilization (access to healthcare facilities, frequency of medical visits), Psychosocial factors (stress, social support), Environmental factors (urban vs. rural residence, access to healthy food options).
Data Analysis: Plan how you will analyze the collected data, including statistical methods to identify independent correlates of glycemic control. Decide on the method for measuring glycemic control, such as HbA1c levels, fasting blood glucose levels, or continuous glucose monitoring. Here HbA1C was used. Use multivariable regression analysis or other appropriate statistical techniques to identify independent correlates of glycemic control while controlling for potential confounding variables.
Interpretation and Reporting: Interpret the findings within the context of existing literature and report the results accurately and transparently in a scientific manuscript or report. Here the authors only formulate tables, however, addition of some appropriate figure/map in methodology section could make the manuscript more interesting and innovative. Obtain ethical approval from relevant institutional review boards and ensure that participant confidentiality and privacy are maintained throughout the study.
Issues: However, there are some areas for improvement and critique: The manuscript should ensure that any language directly quoted from other sources is properly cited to avoid plagiarism. The manuscript could benefit from a clearer statement of the study's significance. Why is understanding the correlates of glycemic control in South Africa important? How do the findings contribute to existing knowledge or address gaps in the literature? Adding a sentence or two to explicitly state the broader implications of the study could enhance the overall impact of the write up.
While the manuscript provides a clear outline of the study's purpose and findings, some sentences could be more concise. For instance, in the abstract section, "Literature on understanding the drivers of glycemic control in LMICs and Sub-Saharan Africa, where the prevalence of diabetes is high, is sparse" could be condensed without losing meaning.
The manuscript mentioned the use of a "fully adjusted linear regression model" but does not provide specific details about the variables included in the model or how they were adjusted for. Adding a sentence to briefly describe the methodology in more detail could enhance understanding.
It's important to acknowledge any limitations of the study. Although three limitations were provided at the end of the manuscript. However, there could be another limitation , for example, the reliance on self-reported data for variables such as education and income.
While the conclusions mention potential targets for interventions, it could be beneficial to include a brief mention of future research directions. This could involve areas where further investigation is needed or potential implications for policy and practice based on the findings.
Author Response
Reviewer #1:
Review Report: Thank you very much for providing me the opportunity to review this manuscript. While reading the manuscript, I could make the following comment from my side:
Summary: The escalating prevalence of diabetes, particularly in low and middle-income countries like those in Sub-Saharan Africa, underscores the urgent need for research on glycemic control. This analysis aimed to pinpoint the factors influencing glycemic control in South Africa. Utilizing data from the 2016 South Africa Demographic and Health Survey, variables such as age, ethnicity, and wealth index were examined alongside HbA1c levels. Significant disparities in HbA1c were observed across various demographics. This study, encompassing a sizable and diverse population, offers vital insights into enhancing interventions aimed at ameliorating diabetes-related health outcomes in South Africa.
Here are some general suggestions to the authors for conducting such study in future and the authors tried to follow a few of them conducting this research:
Sampling Method and Participant Recruitment: Thank you for the opportunity to provide additional details about the sampling method and study participants. This manuscript is an analysis of international, publicly available secondary data for the country of South Africa from the Demographic and Health Survey study. The following highlighted text was added to section 2.1 “Data Source and Study Population” of the methods on page 2:
This study is a secondary data analysis of publicly available international data that were collected by the Demographic and Health Survey (DHS) program through five surveys conducted in South Africa between June 2016 and November 2016, the most recent year of available data for South Africa. The DHS program collects data on population health from over 100 countries around the world (National Department of Health 2019).
In addition to adding clarifying language describing the dataset, we have responded to each concern raised regarding sampling and sample size below and have also added these details to the methods section of the paper.
- Determine how you will select participants, whether it's through random sampling, stratified sampling, or convenience sampling. The Statistics South Africa Master Sample Frame that was created by using enumeration areas from the 2011 Census was used for the SADHS. South Africa is divided into nine provinces and the SADHS was designed to provide estimates for the entire country of South Africa, as a result primary sampling units were allocated by a power allocation to ensure survey precision across provinces, and each province was stratified into urban, farm, and traditional areas resulting in 26 sampling strata. The SADHS used a stratified two-stage sample design with stage 1 using a probability that was proportional to the size of primary sampling units, and stage 2 of systematic sampling of dwelling units.
The above language was added to section 2.1 “Data Source and Study Population” of the methods in the first paragraph on page 3. In addition, the citation for the SADHS methodology document was added in text and to the reference section of the manuscript. This reference is listed as number 42 in the reference section.
- Develop a plan to recruit participants from various regions in South Africa who have been diagnosed with diabetes. The number of dwelling units identified from the 2011 Census was used as the primary sampling unit measure of size with 750 primary sampling units selected from 26 sampling strata which resulted in 468 primary sampling units in urban areas, 224 primary sampling units in traditional areas, and 58 primary sampling units in farm areas. Lists of dwelling units served as the sampling frame and a fixed number of 20 dwelling units per cluster were selected with systematic selection from the list. Households within selected dwelling units were eligible for interviews and study participation.
This language along with appropriate citations was added to section 2.1 “Data Source and Study Population” of the methods in the second paragraph on page 3.
- For sample size Calculation, determine the required sample size based on the expected effect size, power, and significance level. The SADHS is representative of the national population ages 15-49, and the SADHS is representative at the national and provincial levels for both urban and nonurban areas. Provinces containing small populations were oversampled. For example, the number of women interviewed ranged from 656 in Western Cape to 1360 in KwaZulu-Natal. The total national sample size of women was 8514 before applying our specific study inclusion criteria. Similar methods were also used to sample men.
This language along with appropriate citations was added to section 2.1 “Data Source and Study Population” of the methods in the second paragraph on page 3.
Data Collection: Thank you for the opportunity to provide additional details about the data collection strategies used for SADHS. As previously described, this manuscript is an analysis of international, publicly available secondary data. We have responded to each concern raised regarding data collection below and have also added these details to the methods section of the paper.
- Design a standardized questionnaire or interview protocol to collect relevant data from participants. This could include demographic information, medical history, lifestyle factors, and details about diabetes management. For Data Management, establish a system for data entry, storage, and management to ensure data integrity and security. The questionnaires used for the SADHS were based on the Demographic and Health Survey Program’s standard DHS questionnaires and were adapted to reflect the population and health issues that were relevant to South Africa. This manuscript analyzes data that was collected using these standardized questionnaires.
The following sentences were added to the 4th paragraph of the methods section found on page 3.
“The questionnaires used for the SADHS were based on the Demographic and Health Survey Program’s standard DHS questionnaires and were adapted to reflect the population and health issues that were relevant to South Africa.”
This study only includes adult women and men with diabetes in South Africa, and reports on the findings of an analysis that was conducted using previously collected SADHS data from 2016.
- Implement measures to ensure the quality and reliability of data collection, such as training interviewers and conducting pilot studies. Training strategies used by the SADHS team include classroom training and field practice activities. About 300 field workers were recruited and trained. Field workers included interviewers, supervisors, logistics officers, and nurses. The training course included instructions on interviewing techniques and administration of paper and electronic questionnaires, field procedures, questionnaire content review, and mock interviews. Nurses were trained on how to collect biomarker data. More detailed information on the training procedures can be found in the SADHS Survey Methods manual.
The following language was added to the 3rd paragraph of the methods found on page 3:
Training strategies used by the SADHS team included classroom training and field practice activities (National Department of Health, 2019). About 300 field workers (interviewers, supervisors, logistics officers, and nurses) were recruited and trained. The training course included instructions on interviewing techniques and administration of paper and electronic questionnaires, field procedures, questionnaire content review, and mock interviews (National Department of Health, 2019). More detailed information on the training procedures can be found in the SADHS Survey Methods manual (National Department of Health, 2019).
Dependent Variable:
- Glycemic control level (e.g., HbA1c levels or fasting blood glucose). The dependent variable for the analysis, glycemic control, was defined as hemoglobin A1C. This information can be found in section 2.2 “Dependent Variable and Correlates” of the Methods section of the manuscript on page 4.
Independent Variables:
- Demographic factors (age, gender, ethnicity, socioeconomic status), Clinical factors (duration of diabetes, comorbidities, medication regimen), Lifestyle factors (diet, physical activity, smoking, alcohol consumption), Healthcare access and utilization (access to healthcare facilities, frequency of medical visits), Psychosocial factors (stress, social support), Environmental factors (urban vs. rural residence, access to healthy food options). We appreciate this thoughtful consideration; however, we were interested in identifying and understanding the socioeconomic correlates of diabetes in this country and variables were selected based on this premise and based on the socioeconomic variables that were available in this secondary dataset. As such, the following variables from your list were included in the analysis: Demographic factors (age, gender, ethnicity, marital status, and socioeconomic status [ability to read, educational level, wealth index, occupation group, and employment during the last 12 months]), healthcare access (insurance), and environmental factors (region, place of residence).
The following language was added to section 2.3 “Statistical Analysis” section of the methods on page 4: A fully adjusted linear regression model that included glycemic control as measured by continuous HbA1c as the outcome and the following independent variables was used to identify the independent correlates of glycemic control: age, gender, ethnicity, marital status, region, place of residence, able to fully read, educational level, insurance, wealth index, occupation group, and employment during the last 12 months. As the primary goal of the paper was to understand the independent socioeconomic correlates of glycemic control, variables were selected based primarily on the predisposing and enabling con-structs of the Andersen model of health services utilization and outcomes research (Andersen, 1995). In addition, the Andersen citation was added as the 43rd reference in the references list as:
- Andersen R. (1995). Revisiting the behavioral model and access to medical care: Does it matter? Journal of Health and Social Behavior, 36(1): 1- 10.
Data Analysis:
- Plan how you will analyze the collected data, including statistical methods to identify independent correlates of glycemic control. We appreciate this recommendation. Section 2.3 “Statistical Analysis” of the methods section describes the analyses that were done to identify the independent correlates of glycemic control as measured using hemoglobin A1C. We have added the list of independent variables that were added to the fully adjusted linear regression model in this section to clarify which variables were considered in the analysis.
- Decide on the method for measuring glycemic control, such as HbA1c levels, fasting blood glucose levels, or continuous glucose monitoring. Here HbA1C was used. Yes, we used hemoglobin A1C for glycemic control.
- Use multivariable regression analysis or other appropriate statistical techniques to identify independent correlates of glycemic control while controlling for potential confounding variables. We appreciate the opportunity to clarify the methods used to identify independent correlates while controlling for potential confounding variables. Yes, we used multivariable linear regression to identify independent correlates of glycemic control. As mentioned above, we have added these details to the statistical analysis section of the methods.
Interpretation and Reporting:
- Interpret the findings within the context of existing literature and report the results accurately and transparently in a scientific manuscript or report. Here the authors only formulate tables, however, addition of some appropriate figure/map in methodology section could make the manuscript more interesting and innovative. We appreciate this feedback and have made the following edits to address the concerns raised. First, we have added the SADHS map of South Africa that illustrates the provinces included in the SADHS data that were collected to section 2.1 “Data Source and Study Population” of the methods and have listed the provinces included in the study. These additions can be found on page 3.
Second, we have included written text description of the findings in the Results section of the manuscript, found on page 5.
- Obtain ethical approval from relevant institutional review boards and ensure that participant confidentiality and privacy are maintained throughout the study. We appreciate this thoughtful consideration and have added information regarding the ethics approval to the methods section of the manuscript. The SADHS survey protocol was reviewed and approved by the South African Medical Research Council and Inner City Fund Institutional Review Board.
Issues: However, there are some areas for improvement and critique:
- The manuscript should ensure that any language directly quoted from other sources is properly cited to avoid plagiarism. The manuscript has been reviewed and appropriate citations have been included to avoid plagiarism.
- The manuscript could benefit from a clearer statement of the study's significance. Why is understanding the correlates of glycemic control in South Africa important? How do the findings contribute to existing knowledge or address gaps in the literature? Adding a sentence or two to explicitly state the broader implications of the study could enhance the overall impact of the write up. We thank the reviewer for this feedback. This study provides the following primary additions to the field: First, the data on glycemic control as measured by hemoglobin a1c is limited in Sub-Saharan Africa. Given diabetes is under-recognized and under diagnosed in a region of the world where exponential increases of the older adult population are expected to occur in the next 2 – 3 decades, and with this increase in older adult population an increase in noncommunicable diseases such as diabetes; it is imperative that these study findings that identify socioeconomic factors that can serve as potential targets for interventions to improve glycemic control be disseminated within international research, clinical, and political communities. Findings such as those identified in this study can be used to increase the development and testing of efficacious and effective interventions that will improve the health of individuals with diabetes.
Second, conducting this study with data from South Africa, a country with a large diverse population and high prevalence of diabetes provided a unique opportunity to understand the independent correlates of diabetes within an ethnically diverse low- and middle-income country (LMIC).
Third, findings from the study provide information on factors that are associated with increasing hemoglobin a1c, once again affording an opportunity to intervene earlier and before glycemic control reaches poor levels and increases one’s risk of diabetes related complications in under resourced areas that are not equipped to manage complications and complex cases.
Fourth, results from this analysis illustrated disparities in glycemic control by race and ethnicity and the grave need for interventions for Black, Colored, and Indian/Asian adults with diabetes.
These four salient points have been added to the discussion in the second paragraph on page 10.
- While the manuscript provides a clear outline of the study's purpose and findings, some sentences could be more concise. For instance, in the abstract section, "Literature on understanding the drivers of glycemic control in LMICs and Sub-Saharan Africa, where the prevalence of diabetes is high, is sparse" could be condensed without losing meaning. Thank you for this feedback. We have revised the language in this sentence of the abstract to read as, “However, the independent socioeconomic correlates of glycemic control as measured by hemoglobin A1C have yet to be identified.”
We have also thoroughly reviewed the manuscript and have used more concise language where appropriate.
- The manuscript mentioned the use of a "fully adjusted linear regression model" but does not provide specific details about the variables included in the model or how they were adjusted for. Adding a sentence to briefly describe the methodology in more detail could enhance understanding. We appreciate this feedback and have revised the model descriptions to read as follows: A fully adjusted linear regression model was used to identify the independent correlates of glycemic control as measured using continuous HbA1c. The fully adjusted linear regression model included continuous HbA1c as the dependent variable and the following independent variables: age, gender, ethnicity, educational level, marital status, region, place of residence, insurance, ability to read, wealth index, occupation group, and worked in the last 12 months. As the focus of the paper was to understand the independent socioeconomic correlates of glycemic control, variables were selected based primarily on the predisposing and enabling constructs of the Andersen model of health services utilization and outcomes research.
The following citation was added as the 43rd reference and is listed as follows:
- Andersen R. (1995). Revisiting the behavioral model and access to medical care: Does it matter? Journal of Health and Social Behav-ior, 36(1): 1- 10.
- It's important to acknowledge any limitations of the study. Although three limitations were provided at the end of the manuscript. However, there could be another limitation , for example, the reliance on self-reported data for variables such as education and income. We agree with this suggestion and have included a fourth limitation of the reliance on self-reported data for variables such as education and income. The following limitation has been added to the last paragraph of the discussion on page 10: Fourth, the analysis relied on self-reported measures of education and wealth, and while these variables may be subject to recall bias, self-reported measures of education and wealth are commonly used and widely accepted in health services research.
- While the conclusions mention potential targets for interventions, it could be beneficial to include a brief mention of future research directions. This could involve areas where further investigation is needed or potential implications for policy and practice based on the findings. We thank the reviewer for this feedback. We have added the additional study significance and implications content described in comment #22 to the manuscript.

Reviewer 2 Report
Comments and Suggestions for Authors
the manuscript presents an insightful analysis on the determinants of glycemic control among adults with diabetes in South Africa. The study's strength lies in its utilization of a nationally representative dataset, which enhances the generalizability of the findings to the South African population. The selection of a broad range of demographic and socio-economic variables as potential correlates offers a comprehensive approach to understanding the factors influencing glycemic control.
One of the major drawback i see is the old data presentation, why author took so many year to present it.
While the methods summarize the approach well, a more detailed explanation of the statistical methods, including the rationale behind the selection of the fully adjusted linear regression model and its assumptions, would be beneficial. A more thorough discussion of the study's limitations, including any potential biases, the cross-sectional nature of the data, and how these might affect the interpretation and applicability of the findings, would provide a more balanced view. Author should write and suggest how these findings could inform public health strategies and interventions aimed at improving diabetes management and outcomes in South Africa.
Comments on the Quality of English LanguageEnglish is ok. some grammatical and punctuations can be corrected through the manuscript.
Author Response
Reviewer #2:
The manuscript presents an insightful analysis on the determinants of glycemic control among adults with diabetes in South Africa. The study's strength lies in its utilization of a nationally representative dataset, which enhances the generalizability of the findings to the South African population. The selection of a broad range of demographic and socio-economic variables as potential correlates offers a comprehensive approach to understanding the factors influencing glycemic control.
- One of the major drawback i see is the old data presentation, why author took so many year to present it. We appreciate this feedback, however 2016 is the most recent DHS data available for South Africa. We have added this information to the methods to clarify why we used data from 2016. This edit can be found in the 1st paragraph of the methods on page 2.
- While the methods summarize the approach well, a more detailed explanation of the statistical methods, including the rationale behind the selection of the fully adjusted linear regression model and its assumptions, would be beneficial. We thank the reviewer for this feedback. As reviewer 1 raised a similar concern we have included the response from above here as well.
We appreciate this feedback and have revised the model descriptions to read as follows: A fully adjusted linear regression model was used to identify the independent correlates of glycemic control as measured using continuous HbA1c. The fully adjusted linear regression model included continuous HbA1c as the dependent variable and the following independent variables: age, gender, ethnicity, educational level, marital status, region, place of residence, insurance, ability to read, wealth index, occupation group, and worked in the last 12 months. As the focus of the paper was to understand the independent socioeconomic correlates of glycemic control, variables were selected based primarily on the predisposing and enabling constructs of the Andersen model of health services utilization and outcomes research. These edits can be found in the last paragraph of the methods on page 4.
- A more thorough discussion of the study's limitations, including any potential biases, the cross-sectional nature of the data, and how these might affect the interpretation and applicability of the findings, would provide a more balanced view. We appreciate this feedback and have provided a more thorough discussion of the study’s limitations including potential biases and the cross-sectional nature of the data. These edits can be found on page 10.
- Author should write and suggest how these findings could inform public health strategies and interventions aimed at improving diabetes management and outcomes in South Africa. As a similar concern was raised by reviewer 1, we have added the following language to clarify and further highlight how the findings can inform public health strategies and interventions aimed at improving diabetes management outcomes in South Africa. These edits can be found on page 10. The 4th paragraph of the discussion now reads as follows:
This study provides new evidence on the correlates of glycemic control for South Africa, a LMIC in Sub-Saharan Africa, using the biological measure of HbA1c. Given diabetes is under-recognized and under diagnosed in a region of the world where exponential increases of the older adult population are expected to occur in the next 2 – 3 decades, and with it an increase in noncommunicable diseases such as diabetes, it is imperative that findings such as those identified in this paper that can serve as potential targets of interventions designed to improve glycemic control be disseminated with the international research, clinical, and political communities. This new knowledge should be used to increase the development, testing, and funding of efficacious and effective interventions that will improve the health of individuals with diabetes. The findings of this study provide important clinical, research, and policy implications given this new in-formation on the independent correlates of glycemic control among adults with diabetes in South Africa. Conducting this study with data from South Africa, a country with a large diverse population and high prevalence of diabetes, provided a unique opportunity to understand the independent correlates of diabetes within an ethnically diverse low- and middle-income country (LMIC). In addition, study findings highlighted the ethnic dis-parities in glycemic control and illustrated the grave need for interventions for Black, Colored, and Indian/Asian adults with diabetes. Clinical implications of these study findings suggest healthcare providers of adults with diabetes in South Africa should focus on delivering additional diabetes education and continuous skill training for older adults, those in the middle and richest wealth group, and for Black Africans, Colored adults, and Indian/Asian adults. Findings from the study provide information on factors that are associated with increasing hemoglobin a1c, once again affording an opportunity to intervene earlier and before glycemic control reaches poor levels and increasing one’s risk of diabetes related complications in under resourced areas that are not equip to manage complications and complex cases. Therefore, the research implications highlight the importance of developing and testing novel interventions that target the cultural and lifestyle needs of adults in these groups. Finally, political implications include the need for increased access to equitable health care systems and dismantling of systemic factors perpetuating the negative impact of apartheid on socioeconomic status and health.
- Comments on the Quality of English Language: English is ok. some grammatical and punctuations can be corrected through the manuscript. We thank the reviewer for this feedback and have reviewed and revised the grammatical and punctuation errors throughout the paper.

Reviewer 3 Report
Comments and Suggestions for Authors
This is a well-written paper based on a large population-based cohort. The data analyses are appropriate.
It would be interesting to see the results stratified by prior history of diabetes: Yes/No. People who have known diabetes may have different factors associated with good/poor glycaemic control compared with factors in those who are screen-detected.
Author Response
Reviewer #3:
This is a well-written paper based on a large population-based cohort. The data analyses are appropriate.
- It would be interesting to see the results stratified by prior history of diabetes: Yes/No. People who have known diabetes may have different factors associated with good/poor glycaemic control compared with factors in those who are screen-detected. We appreciate this feedback and while we agree this would be interesting, the aim of the paper was to include all adults with diabetes both recognized and unrecognized. Given South Africa is an under resourced area where minimal diabetes-related interventions are being delivered we believed this to be an appropriate starting point for intervention development and dissemination of clinical, research, and political implications. Researchers will be able to work on developing comprehensive strategies to target identified socioeconomic correlates of glycemic control regardless of whether one has diagnosed or undiagnosed diabetes. As more resources are allocated to addressing this disease in LMICs such as South Africa, tailoring interventions based on diagnosed/undiagnosed status may be developed and tested in the context of this environment.
